# An Adaptive Three-Axis Attitude Estimation Method Based on Multi-Sensor Fusion for Optoelectronic Platform

Yawen Kong [1,2], Dapeng Tian [1,2] and Yutang Wang [1,2,*]

1   Key Laboratory of Airborne Optical Imaging and Measurement, Changchun Institute of Optics, Fine Mechanics and Physics, Chinese Academy of Sciences, Changchun 130033, China; kongyawen_0515@163.com (Y.K.); d.tian@ciomp.ac.cn (D.T.)
2   University of Chinese Academy of Sciences, Beijing 100049, China
*   Correspondence: ytwang@ciomp.ac.cn; Tel.: +86-135-7876-5060

**Abstract:** Optoelectronic platform is an important payload in the field of aerospace and widely used in geographic mapping, measurement, and positioning. In order to obtain high-precision attitude measurement, gyro and accelerometer are applied in the feedback loop of light of sight (LOS) control system of optoelectronic platform. Aiming at compensating for gyro drift and maneuvering acceleration disturbance, an adaptive 3-axis attitude estimation method is proposed in this paper. An adaptive threshold criterion is designed by applying the accelerometer data in the sliding window. The threshold is determined in real time to judge whether the maneuvering acceleration exists. If it exists, the angular attitude error is compensated for by the gyro drift model. Otherwise, the angular attitude error is compensated by multi-sensor fusion. Furthermore, a phase-lag-free low pass filter (LPF) is applied to compensate for the phase lag error introduced in the above attitude estimation process. Compared with the angular attitude calculated by gyro, the root mean square error (RMSE) of the proposed method in roll, pitch, and yaw attitude decreased 44.23%, 49.91%, and 46.21%, respectively. In addition, the proposed method can estimate the attitude accurately without obvious phase lag when the maneuvering acceleration disturbance exists. The focus of this paper is to improve the performance of LOS motion control system of optoelectronic platform from the perspective of sensor signal processing. This method is suitable for aerospace applications with high-precision measurement and positioning requirements without maneuver interference, drift error and phase lag.

**Keywords:** optoelectronic platform; adaptive threshold criterion; multi-sensor fusion; attitude estimation

## 1. Introduction

Optoelectronic platform has a wide range of applications in the field of aerospace, such as disaster prevention and rescue, geographic mapping, and precision agriculture [1,2]. The line of sight (LOS) is the axis of the optoelectronic sensor lens. The stability of LOS of optoelectronic sensor can ensure the quality of the output image of optoelectronic platform. A high-quality feedback inertial signal is the basis of LOS motion control system in optoelectronic platform [3]. The processing of inertial sensor signal which affects the stabilization of LOS is an effective method. Gyro and accelerometer are two common inertial space motion sensors [4]. The LOS stability control of the optoelectronic platform needs the inertial velocity feedback of gyro. Gyro has a fast response but suffers from drift due to accumulated errors. In the application of the motion control method, even if the control command and feedback error are zero, the LOS of optoelectronic still shakes due to gyro drift [5]. Therefore, the velocity feedback of gyro measurement must be transformed into attitude feedback. In addition, maneuvering acceleration exists in the flight of aircraft. It adversely affects the output of the accelerometer and the angular attitude calculated

from it [6]. Therefore, the attitude estimation based on multi-sensor data fusion method is necessary to improve the attitude control performance of optoelectronic platform.

Many scholars do studies to solve the above problems. An attitude estimation algorithm based on gyro and accelerometer is proposed in reference [7]. However, the cumulative error of yaw attitude cannot be compensated for due to the lack of magnetometer information. An error data model of accelerometer and gyro is established and the attitude of moving carrier is predicted by Kalman filter [8,9]. This method can suppress the attitude divergence caused by gyro drift but it is not applicable to the situation where the motion state changes rapidly. The system has poor robustness because the accelerometer measures the sum of motion acceleration and gravity acceleration when the carrier has maneuvering acceleration [10]. An extended Kalman filter (EKF) algorithm is applied to integrate accelerometer and magnetometer in [11]. During the maneuvering period, the estimation error of the maneuvering acceleration is modeled as the sum of a fluctuation error and a sudden change error. However, the magnetometer has measurement error due to electromagnetic interference which affects the measurement accuracy [12]. An improved adaptive Kalman filter (AKF) based on Sage-Husa time-varying state disturbance estimator is used to estimate the mean and variance of state disturbance, but this method is not stable and easy to diverge [13].The combination of KF and adaptive Kalman filter is also an effective data fusion method [14]. Reference [15] takes GPS/SINS integrated navigation as the background and adopts adaptive filter algorithm for attitude estimation. However, the measurement error of GPS is large when obstructions exist which affects the accuracy of the algorithm [16]. In addition, it needs to be processed synchronously with inertial sensor which increases the complexity of the algorithm.

In order to solve the above problems, zero speed correction and speed constraint are applied to improve the accuracy and robustness of attitude estimation algorithm [17]. It means that the accurate detection of motion state of the moving carrier is very important. Reference [18] uses accelerometer variance and introduces yaw attitude for detection. The motion state is judged by the standard deviation (STD) of *X*-axis and *Y*-axis data of accelerometer and magnetometer [19]. These methods are mainly used in vehicle navigation system but there is little research on attitude measurement. In addition, it is difficult to meet the real-time and flexibility requirements of LOS stabilization control system of the optoelectronic platform.

Furthermore, dynamic processes lead to phase lag. If the feedback signal lags behind, the control system is difficult to achieve a high performance [20]. The purpose of introducing additional signal is reducing the phase lag. Low pass filter (LPF) and high pass filter (HPF) can form a filter to utilize the phase advance information [21]. However, it is difficult for any input signal to compensate for the phase lag of LPF. A phase-lag-free LPF has been proposed using higher-order signal in [22]. This method can suppress noise without obvious phase lag.

Although the common data fusion method can effectively suppress the gyro drift, the measurement accuracy is not high due to the influence of maneuver. The application of GPS and magnetometer can suppress the maneuvering acceleration interference effectively. However, GPS is not suitable for suspended platform without movement since it is necessary to establish the correction quantity related to the inertial attitude [23]. The measurement accuracy of magnetometer is not high due to electromagnetic interference [12]. In addition, signal filtering will lead to phase lag inevitably. Compensating the phase lag is also effective to improve the performance of the control system. Optoelectronic platforms adopt vibration isolators to isolate the interference from the carrier, such as attitude motion and vibration [24]. The LOS stability control accuracy of optoelectronic platform can be improved as long as the attitude estimation based on multi-sensor fusion is applied. The contributions of this study are as follows:

(a) An adaptive threshold criterion is designed based on the accelerometer data in the sliding window. The threshold is determined online to judge whether the influence of maneuvering acceleration can be ignored;

(b) Data fusion based on gyro and accelerometer (DFBGA) and suppressing maneuvering acceleration disturbance method (SLAIM) are designed in attitude estimation. Based on adaptive threshold criterion, DFBGA is applied if the influence of maneuvering acceleration can be ignored. SLAIM is applied if the influence of maneuvering acceleration cannot be ignored. It is also adopted in the estimation of yaw attitude because it cannot be measured by accelerometer;

(c) In order to compensate for the phase lag error introduced in the above attitude estimation process based on Kalman filter, a phase-lag-free LPF is proposed to revise the phase lag by applying the higher order signal.

This paper is organized as follows. The problem formulation is introduced in Section 2. The attitude estimation method design is proposed in Section 3. Simulations and experiments are described in Section 4. At last, Section 5 concludes the paper.

## 2. Problem Formulation

Body coordinate system and geographic coordinate system should be established as the reference coordinate system for attitude change. The coordinate system determined by the body axis of the carrier is the body coordinate system $O_b - X_b Y_b Z_b$ and the coordinate system determined by east-north-sky direction is geographic coordinate system $O_g - X_g Y_g Z_g$. These two coordinate systems are shown in Figure 1. The accelerometer can obtain linear acceleration and the gyro can obtain angular velocity in the body coordinate system. The angular attitude and angular velocity in the geographic coordinate system can be obtained by solving the output data of these sensors. Roll angle $\gamma$ is the angular attitude rotating around $X_g$-axis, pitch angle $\theta$ is the angular attitude rotating around $Y_g$-axis and yaw angle $\psi$ is the angular attitude rotating around $Z_g$-axis.

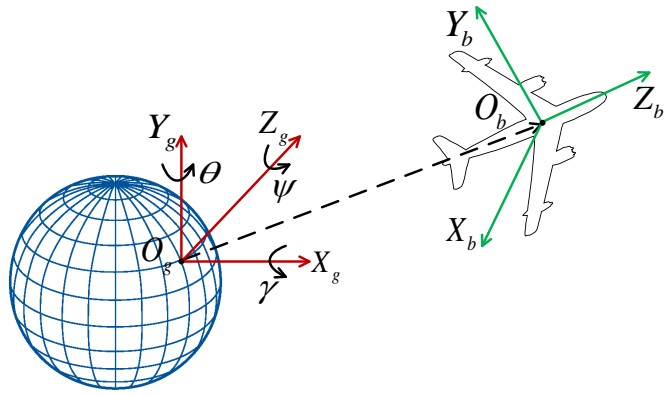

**Figure 1.** Coordinate system setting.

The measured value of gyro is the angular velocity of body coordinate system. It has the advantage of fast response and low cost. However, it suffers from drift due to accumulated errors by integration to obtain angle information. The measured value of accelerometer is the linear acceleration of body coordinate system which does not suffer from zero drift. However, the yaw angle $\psi$ cannot be measured by accelerometer due to the rotation plane of $\psi$ is orthogonal to gravity. Therefore, it is necessary to use the signal processing method of fusing multiple sensor data to improve the accuracy of angular attitude measurement.

## 3. An Adaptive Attitude Estimation Method Based on Multi-Sensor Fusion

In this study, attitude estimation algorithm flow chart is shown in Figure 2. It includes four parts: coordinate transformation, adaptive threshold setting, attitude measurement based on Kalman filter, and phase-lag-free LPF.

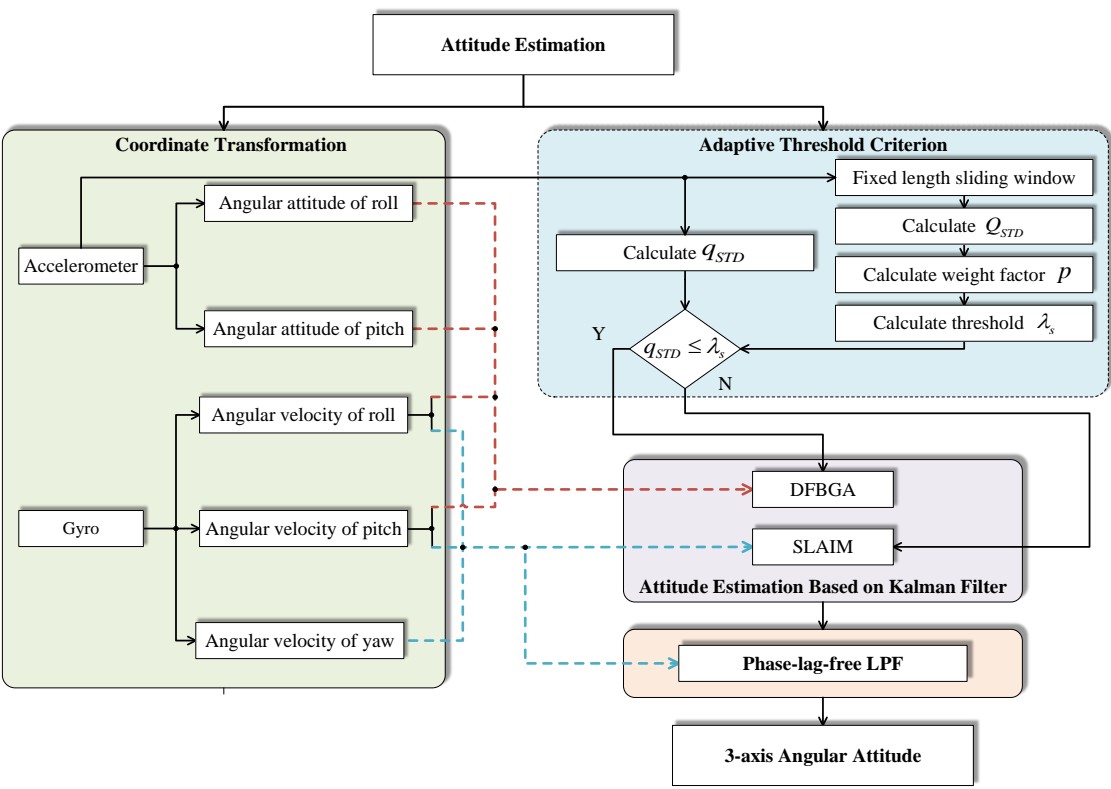

**Figure 2.** Attitude estimation algorithm flow chart.

### 3.1. Coordinate Transformation

The coordinate transformation for accelerometer output data is shown as follows. If an vector in the inertial space is calculated by unit vector $[0\ 0\ 1]^T$ rotating around 3-axis, attitude rotation matrix $C_b^g$ can be expressed as

$$C_b^g = \begin{bmatrix} \cos\gamma\cos\psi & \cos\psi\sin\gamma\sin\theta - \cos\theta\sin\gamma & \sin\theta\sin\psi + \cos\theta\cos\psi\sin\gamma \\ \cos\gamma\sin\psi & \cos\theta\cos\psi + \sin\gamma\sin\theta\sin\psi & \cos\theta\sin\gamma\sin\psi - \cos\psi\sin\theta \\ -\sin\gamma & \cos\gamma\sin\theta & \cos\gamma\cos\theta \end{bmatrix}. \tag{1}$$

Record the accelerometer measurement as $a_x$, $a_y$, and $a_z$ after unitization

$$\begin{bmatrix} a_x \\ a_y \\ a_z \end{bmatrix} = C_b^g \begin{bmatrix} 0 \\ 0 \\ 1 \end{bmatrix} = \begin{bmatrix} \sin\theta\sin\psi + \cos\theta\cos\psi\sin\gamma \\ \cos\theta\sin\gamma\sin\psi - \cos\psi\sin\theta \\ \cos\gamma\cos\theta \end{bmatrix}. \tag{2}$$

In Equation (2), when the yaw angle $\psi = 0$, the roll angle $\gamma$, and pitch angle $\theta$ can be calculated as Equations (3) and (4). However, the yaw angle cannot be measured by accelerometer due to the rotation plane is orthogonal to gravity.

$$\gamma = -\arcsin a_x, \tag{3}$$

$$\theta = \arctan\left(\frac{a_y}{a_z}\right). \tag{4}$$

The coordinate transformation of gyro output data is shown as follows. Its 3-axis angular velocity is $\omega_x$, $\omega_y$ and $\omega_z$,

$$\begin{bmatrix} \omega_x \\ \omega_y \\ \omega_z \end{bmatrix} = \begin{bmatrix} \cos\theta\cos\gamma & \cos\gamma & 0 \\ -\sin\theta & 0 & 1 \\ -\cos\theta\cos\gamma & \sin\gamma & 0 \end{bmatrix} \begin{bmatrix} \dot{\theta} \\ \dot{\gamma} \\ -\dot{\psi} \end{bmatrix}. \tag{5}$$

Therefore, 3-axis angular velocity in the geographic coordinate system can be obtained by

$$
\begin{bmatrix} \dot{\psi} \\ \dot{\theta} \\ \dot{\gamma} \end{bmatrix} = \begin{bmatrix} \frac{\sin\gamma}{\cos\theta} & 0 & -\frac{\cos\gamma}{\cos\theta} \\ \cos\gamma & 0 & \sin\gamma \\ \sin\gamma\tan\theta & 1 & -\cos\gamma\tan\theta \end{bmatrix} \begin{bmatrix} \omega_x \\ \omega_y \\ \omega_z \end{bmatrix}. \tag{6}
$$

### 3.2. Adaptive Threshold Criterion

The accelerometer output data are used as the basis to judge the influence of maneuvering acceleration for the attitude estimation of roll angle $\gamma$ and pitch angle $\theta$, respectively. Collect the output data of $X$-axis and $Y$-axis of accelerometer in the conditions of the maneuvering acceleration exists and does not exist. If there are $N$ groups of data within the sampling interval, the standard deviation(STD) of them can be expressed as

$$
q_{STD} = \sqrt{\frac{1}{N-1}\sum_{i=1}^{N}(A_i - \mu)^2}, \tag{7}
$$

where $A_i$ is the $i$-th output of the single axis of the accelerometer, and $\mu$ is the average value of $N$ groups data of the corresponding axis.

Take the $X$-axis as an example and collect the output data of $X$-axis of accelerometer in two conditions of the same interval. The STD of two conditions are shown in Figure 3. It shows that the STD of acceleration changes slowly when the maneuvering acceleration does not exist and the STD of the acceleration fluctuates greatly and changes rapidly when the maneuvering acceleration exists.

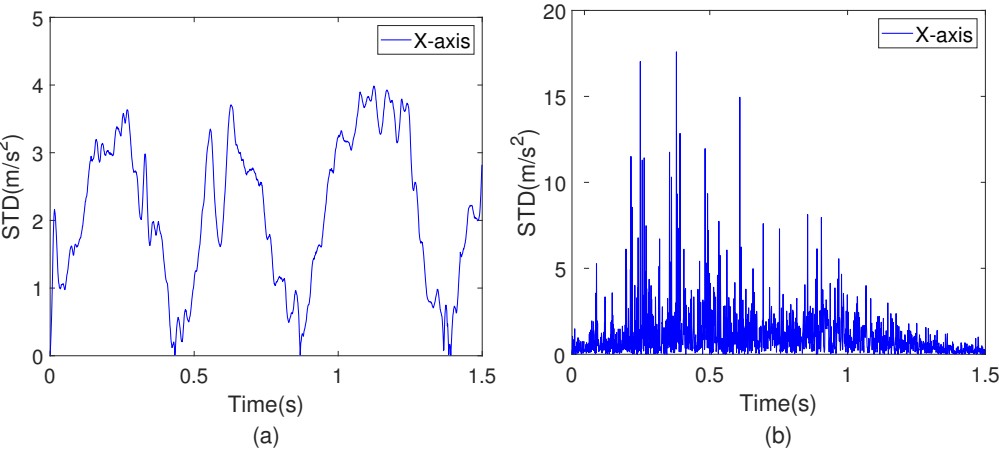

**Figure 3.** STD of the output of X-axis accelerometer. (**a**) maneuvering acceleration does not exist. (**b**) maneuvering acceleration exists.

Based on the above analysis, an adaptive threshold criterion method is adopted. The threshold value $\lambda_{sx}$ is calculated according to the STD of $x$-axis and $y$-axis acceleration obtained in the fixed length sliding window, which can be expressed

$$
\lambda_{sx} = \sum_{i=1}^{n}(Q_{STDxi} \cdot p_{xi}) \cdot k_x, \tag{8}
$$

where $n$ is the fixed window length which used to calculate the threshold. $k_x$ is the magnification factor. $Q_{STDxi}$ is the STD value calculated of each sampling interval in the window time. $p_{xi}$ is the weight factor and $p_{x1} \le p_{x2} \le \cdots \le p_{xn}$.

$$
p_{xi} = \frac{i^m}{\sum_{i=1}^{n} i^m}, m = 2, \tag{9}
$$

where *i* is the serial number in the window time. Compared with $m = 1$, it can increase the influence of approaching time; compared with $m > 2$, it will not highlight the approaching time too much to suppress the influence of gross error on the threshold calculation.

If the STD of acceleration calculated in the current period is less than the threshold $\lambda_{sx}$, it could be considered that the influence of the maneuvering acceleration in the current period can be ignored. Otherwise, it is considered that the influence of the acceleration in the current period cannot be ignored. The threshold setting and the method of judging whether the maneuvering acceleration exists of *Y*-axis are similar to that of *X*-axis.

### 3.3. Attitude Estimation Based on Kalman Filter

The accuracy of the angular attitude calculated by the accelerometer is influenced by the maneuvering acceleration in motion. In addition, the yaw angle cannot be calculated by the output data of the accelerometer. Therefore, two attitude estimation algorithms are designed based on Kalman filter.

For the roll angle $\gamma$ and pitch angle $\theta$, if the STD of the acceleration output data calculated in the current period is greater than the threshold of the corresponding axis, that is the influence of maneuvering acceleration cannot be ignored and for the yaw angle $\psi$, the angular attitude is calculated by SLAIM. If the STD of the output data of accelerometer calculated in the current period is less than the threshold of the corresponding axis, that is the influence of the maneuvering acceleration can be ignored, the angular attitude is calculated by DFBGA.

#### 3.3.1. Kalman Filter for Inertial Attitude Estimation

Set state equation and measurement equations of random linear discrete system as:

$$\begin{cases} x_k = A_{k,k-1}x_{k-1} + B_{k,k-1}u_k + \omega_k \\ z_k = H_k x_k + v_k \end{cases}, \tag{10}$$

where $x_k$ is the state variable, $z_k$ is the measurement variable, $A_{k,k-1}$ is the non-singular state one step transition matrix, $B_{k,k-1}$ is the process noise input matrix, $u_k$ is the control input, $\omega_k$ is the random process noise sequence, $H_k$ is the measurement matrix and $v_k$ is the measurement noise sequence.

Time update equations:

$$\hat{x}_k^- = A\hat{x}_{k-1}^- + B\hat{u}_{k-1} + \hat{\omega}_{k-1}, \tag{11}$$

$$P_k^- = AP_{k-1}A^T + Q. \tag{12}$$

State update equations:

$$K_k = P_k^- H^T (HP_k^- H^T + R)^{-1}, \tag{13}$$

$$\hat{x}_k = \hat{x}_k^- + K_k(z_k - H\hat{x}_k^-), \tag{14}$$

$$P_k = (I - K_k H)P_k^-. \tag{15}$$

#### 3.3.2. Data Fusion Method of Gyro and Accelerometer

For the angular attitude measurement of roll angle $\gamma$ and pitch angle $\theta$ when the influence of maneuvering acceleration can be ignored, the data of gyro and accelerometer are fused to obtain accurate attitude estimation value by Kalman filter.

The roll angle $\gamma$ and pitch angle $\theta$ are taken as the state variables of Kalman filter, that is the state variables of the state equation in Equation (10) is

$$x_k = \begin{bmatrix} \gamma'_k & \theta'_k \end{bmatrix}^T. \tag{16}$$

The angular attitude calculated by the accelerometer and the angular velocity calculated by the gyro in the geographic coordinate system are taken as the measurement

variables of Kalman filter. The measurement variables of the measurement equation in Equation (10) is

$$z_k = \begin{bmatrix} \gamma_k & \dot{\gamma}_k & \theta_k & \dot{\theta}_k \end{bmatrix}^T, \tag{17}$$

where $\gamma'_k$ and $\theta'_k$ are the actual roll and pitch angular attitude, $\gamma_k$ and $\theta_k$ are the roll and pitch angular attitude calculated by the accelerometer and $\dot{\gamma}_k$ and $\dot{\theta}_k$ are the roll and pitch angular velocity calculated by the gyro at $k$-time. In this fusion method, control input $\mu_k = 0$, random process noise sequence $\omega_k$ and measurement noise sequence $v_k$ can be ignored.

### 3.3.3. Attitude Estimation for Suppressing Maneuvering Acceleration Disturbance

For the angular attitude estimation of roll angle $\gamma$ and pitch angle $\theta$ when the influence of maneuvering acceleration cannot be ignored as well as the angular attitude estimation of yaw angle $\psi$, the attitude error caused by gyro dynamic drift is regarded as a time-varying signal. A more accurate attitude error estimation value is obtained through Kalman filter to compensate for the attitude angle error.

There is an attitude drift error between the angular attitude calculated by the gyro and the actual angular attitude due to the existence of gyro zero drift. The relationship between the actual angular attitude and the calculated angular attitude at moment $k$ is as follows:

$$\begin{bmatrix} \gamma'_k \\ \theta'_k \\ \psi'_k \end{bmatrix} = \begin{bmatrix} \gamma_k - \varepsilon_{\gamma k} \\ \theta_k - \varepsilon_{\theta k} \\ \psi_k - \varepsilon_{\psi k} \end{bmatrix}, \tag{18}$$

where $\gamma'_k$, $\theta'_k$ and $\psi'_k$ are the actual 3-axis angular attitude at moment $k$, $\gamma_k$, $\theta_k$, and $\psi_k$ are the 3-axis angular attitude calculated by the gyro at moment $k$ and $\varepsilon_{\gamma k}$, $\varepsilon_{\theta k}$, and $\varepsilon_{\psi k}$ are the 3-axis angular attitude drift error at moment $k$.

The attitude error caused by the zero drift of gyro is related to the motion complexity of the measured carrier and the attitude error accumulates with time when measuring the moving carrier. The motion complexity is described as the change of the angular attitude at moment $k$ relative to the angular attitude at moment $k-1$ and the attitude drift error model equation is established as

$$\begin{bmatrix} \varepsilon_{\gamma k} \\ \varepsilon_{\theta k} \\ \varepsilon_{\psi k} \end{bmatrix} = \begin{bmatrix} \varepsilon_{\gamma(k-1)} \\ \varepsilon_{\theta(k-1)} \\ \varepsilon_{\psi(k-1)} \end{bmatrix} + \delta_j \cdot \begin{bmatrix} \gamma_k - \gamma_{k-1} \\ \theta_k - \theta_{k-1} \\ \psi_k - \psi_{k-1} \end{bmatrix} + \begin{bmatrix} \omega_{\gamma k} \\ \omega_{\theta k} \\ \omega_{\psi k} \end{bmatrix}, \tag{19}$$

where $\varepsilon_{\gamma k}$, $\varepsilon_{\theta k}$ and $\varepsilon_{\psi k}$ are the 3-axis angular attitude drift error at moment $k$, $\varepsilon_{\gamma(k-1)}$, $\varepsilon_{\theta(k-1)}$ and $\varepsilon_{\psi(k-1)}$ are the 3-axis angular attitude drift error at moment $k-1$ and $\delta_j$, $j = 1, 2, 3$ is adjustable parameter. Random process noise sequence $\omega_{\gamma k}$, $\omega_{\theta k}$ and $\omega_{\psi k}$ are determined by the measurement accuracy of the gyro.

The state equation of Kalman filter is established according to the attitude drift error model. The state variable is

$$x_k = \begin{bmatrix} \varepsilon_{\gamma k} & \varepsilon_{\theta k} & \varepsilon_{\psi k} \end{bmatrix}^T. \tag{20}$$

The control variable is

$$u_k = \begin{bmatrix} \gamma_k - \varepsilon_{\gamma k} & \theta_k - \varepsilon_{\theta k} & \psi_k - \varepsilon_{\psi k} \end{bmatrix}^T. \tag{21}$$

In our study, the LPF between the angular attitude calculated by the gyro at moment $k$ and the first few moments of moment $k$ is used as the measurement of attitude drift error caused by gyro drift. The measurement variable of attitude drift error is

$$z_k = a_j \cdot \begin{bmatrix} \gamma_k \\ \theta_k \\ \psi_k \end{bmatrix} + b_j \cdot \begin{bmatrix} \gamma_{k-1} \\ \theta_{k-1} \\ \psi_{k-1} \end{bmatrix} + c_j \cdot \begin{bmatrix} \gamma_{k-2} \\ \theta_{k-2} \\ \psi_{k-2} \end{bmatrix} - \begin{bmatrix} \gamma_{k-3} \\ \theta_{k-3} \\ \psi_{k-3} \end{bmatrix}, j = 1, 2, 3, \tag{22}$$

where $a_j, b_j, c_j, j = 1, 2, 3$ meet the condition $a_j + b_j + c_j = \begin{bmatrix} 1 & 1 & 1 \end{bmatrix}^T$. $\gamma_k, \gamma_{k-1}, \gamma_{k-2}, \gamma_{k-3}$, $\theta_k, \theta_{k-1}, \theta_{k-2}, \theta_{k-3}, \psi_k, \psi_{k-1}, \psi_{k-2}, \psi_{k-3}$ are the 3-axis angular attitudes calculated by gyro at moment $k, k-1, k-2, k-3$, respectively.

### 3.4. Phase-Lag-Free Low Pass Filter

Phase lag error will be introduced in the above attitude estimation process based on Kalman filter. An additional high-order signal with phase advanced information is introduced to filter the original signal. The phase lag can be compensated for by phase-lag-free LPF.

Define

$$\alpha_k = \begin{bmatrix} \gamma'_k & \theta'_k & \psi'_k \end{bmatrix}^T, \tag{23}$$

$$\beta_k = \begin{bmatrix} \dot{\gamma}_k & \dot{\theta}_k & \dot{\psi}_k \end{bmatrix}^T, \tag{24}$$

where $\gamma'_k, \theta'_k, \psi'_k$ are the 3-axis angular attitude calculated by Kalman filter in Section 3.3 and $\dot{\gamma}_k, \dot{\theta}_k, \dot{\psi}_k$ are the 3-axis angular velocity calculated from gyro in Section 3.1. These two inputs are original signals with noise.

Figure 4 shows the design of this filter. The filter can be described as two sensors which could measure the nearby order information. During data process, $\alpha_k$ should be the position signal when $\beta_k$ is the velocity signal. In addition, if $\alpha_k$ is the velocity signal, $\beta_k$ should be the acceleration signal.

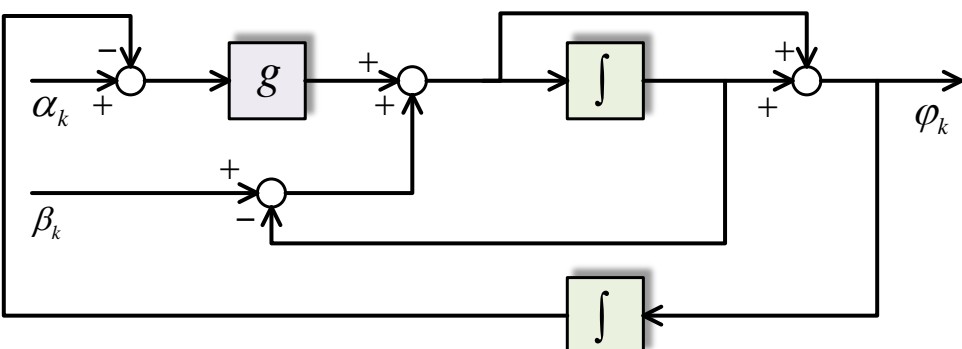

**Figure 4.** Design diagram of phase-lag-free filter.

In this research, the signal $\alpha_k$ is low-pass filtered and the phase lag is corrected by $\beta_k$. The parameter $g$ is cut off frequency of LPF. The tracking characteristic of output signal $\varphi_k$ is strengthened with the increase in $g$. If $\beta_k = \frac{d}{dt}\alpha_k$, the relationship $\varphi_k = \alpha_k$ is met.

Define

$$\begin{cases} \dot{\varphi}_{k1} = g(\alpha_{k1} - \varphi_{k1}) \\ \dot{\varphi}_{k2} = \beta_{k1} - g\varphi_{k2} \end{cases}. \tag{25}$$

The output of system in Figure 4 can be expressed as

$$\varphi_k = \begin{bmatrix} \gamma''_k & \theta''_k & \psi''_k \end{bmatrix}^T = \varphi_{k1} + \varphi_{k2}, \tag{26}$$

where $\gamma''_k, \theta''_k, \psi''_k$ are the 3-axis angular attitude processed by phase-lag-free LPF. It can be seen that the design is composed of LPF and can greatly suppress the input noise from Equation (25).

## 4. Simulations and Experiments

In order to verify the performance of the proposed method in this paper, this section conducts simulations and experiments. Simulations are given in Section 4.1 which includes two parts. In the first part, the attitude estimation method is simulated to verify the accuracy of the method. In the second part, the LOS motion control of optoelectronic platform is simulated to verify the practicability of the method. In Section 4.2, the attitude

estimation method is tested on the embedded system to verify the accuracy and real-time performance of the method.

*4.1. Simulations*

In order to verify the effectiveness of the proposed method, two parts of simulations are carried out in this section. In the first part, the simulated original-gyro signal, the signal processed by DFBGA and the signal processed by proposed method are compared with the reference signal to verify the ability to suppress gyro drift and maneuvering interference. The signal with phase lag and the signal processed by proposed method are compared with the reference signal to verify the ability to suppress phase lag. In the second part, the simulated original-gyro signal, the signal processed by DFBGA and the signal processed by proposed method is applied as the feedback signal of the control system and the position response is compared with the position command. The model parameters of the LOS control system of the optoelectronic platform in [25] are used as a reference.

4.1.1. Simulation Setup

(a) Simulation Setup of Signal Processing

A standard sinusoidal signal $sin(10t)$ is assumed to be an angular attitude reference signal. The random noise $n_a(t) \in [-0.5, 0.5]$ is added to the reference signal as the angular attitude information calculated by the accelerometer and a noise with amplitude $A_a = 1$ is introduced every 0.1 s to simulate the maneuvering acceleration interference. The random noise $n_v(t) \in [-0.1, 0.1]$ is added to the differential of the signal as the angular velocity information calculated by the gyroscope. These two signals are applied in DFBGA and SLAIM. The random noise $n(t) \in [-0.2, 0.2]$ is added to the reference signal and Butterworth filter is carried out to introduce phase lag. This signal is applied in Phase-lag-free LPF. The cut-off frequency $g$ of phase-lag-free LPF is 40 Hz. Other parameters are shown in Tables 1 and 2.

In this simulation, RMSE is applied to compare the signal processed by corresponding algorithm and reference signal.

$$q_{RMSE} = \sqrt{\frac{1}{m} \sum_{k=1}^{m} (y_k - \hat{y}_k)^2},$$

(27)

where $y_k$ is reference signal ,$\hat{y}_k$ is the signal processed by corresponding algorithm and $m$ is number of sampling points.

**Table 1.** Parameters of data fusion method based on gyro and accelerometer in simulation.

| Parameters | Value |
|---|---|
| State transition matrix $A_0$ | 1 |
| Measurement transition matrix $H_0$ | $\begin{bmatrix} 1 & 0 \end{bmatrix}^T$ |
| Process noise $Q_0$ | 0.1 |
| Measurement noise $R_0$ | 0.000001 |

(b) Simulation Setup of Motion Control

This section constructs a simulated single-axis close-loop control of pitch direction for optoelectronic platform, composing of proportion-differentiation (PD) control and disturbance observer (DOB). According to the physical characteristics, the dynamic electromechanical model of each motor can be simplified as

$$J_n \ddot{\vartheta} + B_n \dot{\vartheta} = u + d,$$

(28)

where $J_n$ is nominal mass, $B_n$ is nominal damping, $u$ is control variable, $d$ is system disturbance, $\vartheta$ is angular position response which processed by proposed method and $\dot{\vartheta}$ is angular velocity response which measured by gyro.

**Table 2.** Parameters of attitude estimation for suppressing maneuvering acceleration interference in simulation.

| Parameters | Value |
|---|---|
| State transition matrix $A_0$ | 1 |
| Measurement transition matrix $H_0$ | 1 |
| Process noise $Q_0$ | 0.001 |
| Measurement noise $R_0$ | 0.001 |
| Adjustable parameter $\delta_j$ | 0.3 |
| Random process noise $\omega$ | 0.01 |
| Adjustable parameter $a_j$ | 0.8 |
| Adjustable parameter $b_j$ | 0.1 |
| Adjustable parameter $c_j$ | 0.1 |

The control system diagram is shown in Figure 5. $\vartheta_1$ is the angular position command, $\vartheta_0$ is the angular position response and $\dot{\vartheta}_0$ is the angular velocity response. $d$ is system disturbance and $\hat{d}$ is disturbance estimation. The actual position response is compared with the position command to verify the algorithm.

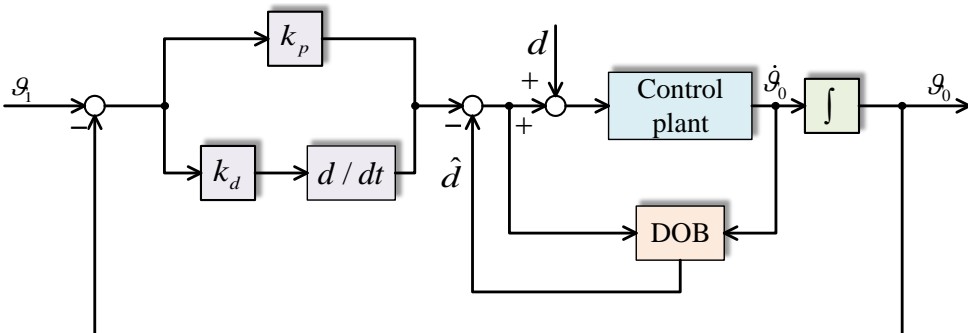

**Figure 5.** Control system diagram.

A standard sinusoidal signal $sin(10t)$ is assumed to be the position command. The motor speed response with random noise $n_v(t) \in [-0.1, 0.1]$ and inertial DC offset which amplitude $A_v = 1$ is used as the simulated original-gyro signal. The motor position response with random noise $n_a(t) \in [-0.5, 0.5]$ is simulated as the angular attitude calculated by accelerometer. A noise with amplitude $A_a = 1$ is introduced every 0.1 s to simulate the maneuvering acceleration interference. The speed response feedback, the feedback signal proposed by DFBGA and the proposed feedback signal are applied in control system. Table 3 shows the parameters of the control system.

**Table 3.** Parameters of control system in simulation.

| Parameters | Value |
|---|---|
| Nominal mass $J_n$ (kg $\cdot$ m$^2$) | 0.0021667 |
| Nominal damping $B_n$ (N $\cdot$ s/m) | 0.15 |
| Proportional gain of PD $K_p$ | 500 |
| Derivative gain of PD $K_d$ | 30 |
| DOB cut-off frequency $g_{dob}$ (Hz) | 200 |

In this simulation, RMSE in Equation (27) is applied to compare the control error between position response and command. Where $y_k$ is position command, $\hat{y}_k$ is position response applying corresponding feedback and $m$ is number of sampling points.

### 4.1.2. Simulation Results of Signal Processing

Figure 6 shows the simulation results of signal process based on Kalman filter. There is obvious noise in the signal processed by DFBGA method due to the simulated maneuvering acceleration. DFBGA and SLAIM method can suppress maneuvering interference effectively. Calculate the RMSE between the signal processed by DFBGA and SLAIM and the reference signal according to Equation (28). Compare with the simulated original-gyro signal, the RMSE of signal processed by DFBGA and SLAIM method decreased 29.08%.

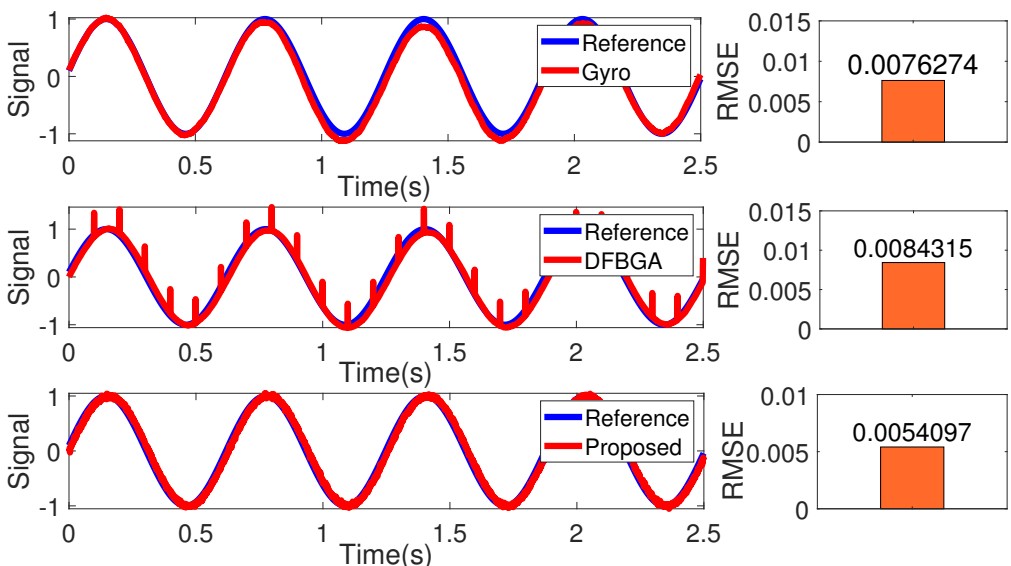

**Figure 6.** Simulation results of signal process based on Kalman filter. The first subfigure is the simulated original-gyro signal, the second subfigure is the signal processed by DFBGA, the third subfigure is the signal processed by DGBGA and SLAIM.

Figure 7 shows the simulation results of signal process based on phase-lag-free LPF. It can be seen from Figure 6 that phase-lag-free LPF can compensate the phase lag effectively. Calculate the RMSE between the signal processed by proposed method and the reference signal according to Equation (27). Compared with the original signal, the RMSE of signal processed by phase-lag-free LPF decreased 98.81%.

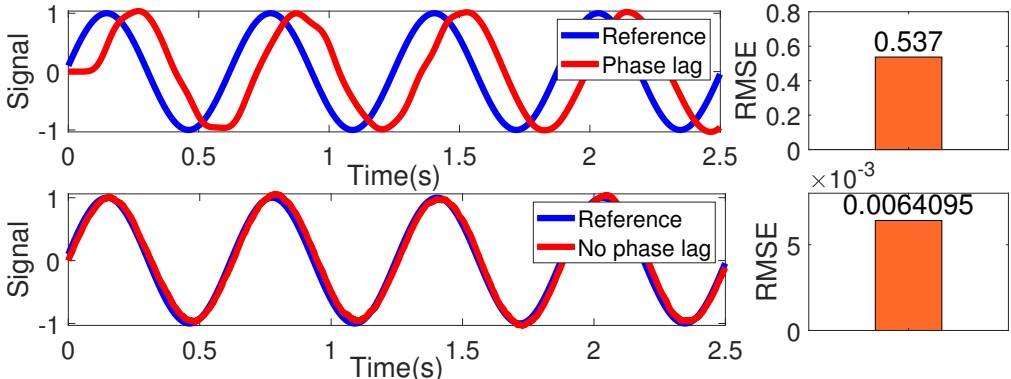

**Figure 7.** Simulation results of signal process based on phase-lag-free LPF. The first subfigure is the original signal with phase lag, the second subfigure is the signal processed by phase-lag-free LPF.

### 4.1.3. Simulation Results of Motion Control

Figure 8 shows the simulation results of motion control which applied feedback signals processed by corresponding methods. Calculate the RMSE between position response and position command according to Equation (27). Compared with the response with gyro feedback, the RMSE of the response with feedback signal via DFBGA decreased 25.51% and the RMSE of the response with feedback signal via proposed method decreased 38.07%. It can be seen that applying the method proposed in this paper to the feedback loop of motion control has more accurate command response and will not cause system instability.

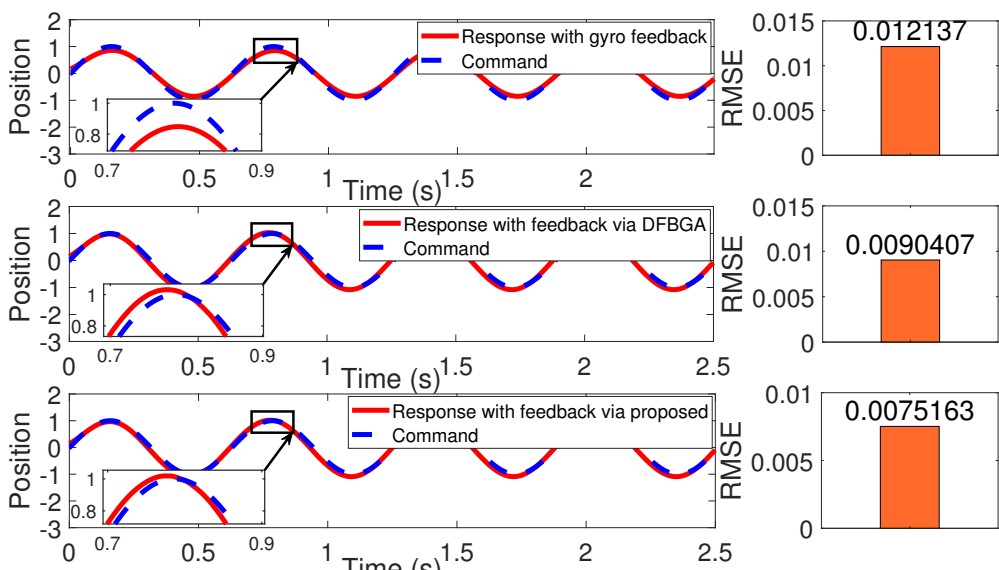

**Figure 8.** Application in motion control. The first subfigure is the simulated original-gyro signal feedback, the second subfigure is the feedback signal processed by DFBGA, the third subfigure is the feedback signal processed by proposed method.

### 4.2. Experiments

#### 4.2.1. Experimental Setup

The experimental device is shown in Figure 9. It is an optoelectronic platform composed of 3-axis servo unit motors with encoders, inertial measurement unit (IMU) with 3-axis gyros and accelerometers, and pay load. The velocity acceleration of the motors are measured by gyros and accelerometers of IMU. The measurement bandwidth and accuracy of encoder can be sure to be convinced for control and the system will not be unstable due to its measurement problems. Therefore, it is reliable to take the encoder measurement data as the standard experimental result. The IMU used in this study is WT931. The accuracy of the gyro is $0.05°/s$ and the accuracy of the accelerometers is $0.1 \text{ m/s}^2$. The encoder applied in this study is AS5048 and its resolution is $0.05°$. The sampling time is 1.5 ms. The cut-off frequency $g$ of phase-lag-free LPF of roll, pitch, and yaw attitude are 150 Hz, 100 Hz, and 150 Hz, respectively. Other parameters are shown in Tables 4 and 5.

In the experiments, the angular attitude calculated by gyro, the angular attitude calculated by DFBGA, the angular attitude calculated by DFBGA and SLAIM, the angular attitude calculated by the proposed method and the angular attitude calculated by 3-axis encoders are compared. This experiment includes two parts. In part 1, the motion carrier moves only with translational motion in the horizontal direction and without 3-axis rotation motion. In part 2, the translational motion in horizontal direction and 3-axis rotation exist in the motion of moving carrier at the same time. The horizontal acceleration is accompanied by slight jitter to simulate the maneuvering motion of UAV. The amplitude of jitter is about 1 cm and the frequency is about 2 to 3 times per second. Due to the limitation of the number of data acquisition software channels, the attitude information of roll, pitch and yaw directions are output, respectively.

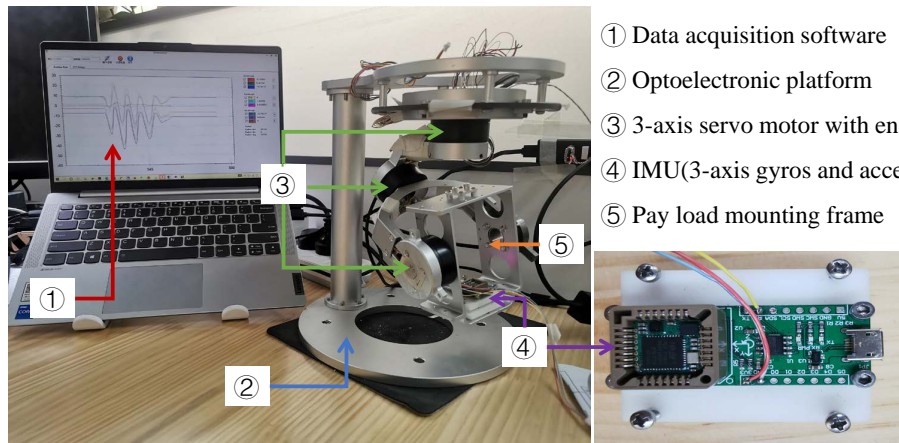

**Figure 9.** Experimental device.

① Data acquisition software
② Optoelectronic platform
③ 3-axis servo motor with encoders
④ IMU(3-axis gyros and accelerometers)
⑤ Pay load mounting frame

**Table 4.** Parameters of data fusion method based on gyro and accelerometer.

| Parameters | Roll | Pitch |
|---|---|---|
| State transition matrix $A_0$ | 1 | 1 |
| Measurement transition matrix $H_0$ | $\begin{bmatrix} 1 & 0 \end{bmatrix}^T$ | $\begin{bmatrix} 1 & 0 \end{bmatrix}^T$ |
| Process noise $Q_0$ (*deg*) | $2.533 \times 10^{-6}$ | $3.1257 \times 10^{-6}$ |
| Measurement noise $R_0$ (*deg*) | $\begin{bmatrix} 2.533 \times 10^{-6} & 0 \\ 0 & 1.4482 \times 10^{-6} \end{bmatrix}$ | $\begin{bmatrix} 3.1257 \times 10^{-6} & 0 \\ 0 & 1.9073 \times 10^{-6} \end{bmatrix}$ |

**Table 5.** Parameters of attitude estimation for suppressing maneuvering acceleration interference.

| Parameters | Roll | Pitch | Yaw |
|---|---|---|---|
| State transition matrix $A_0$ | 1 | 1 | 1 |
| Measurement transition matrix $H_0$ | 1 | 1 | 1 |
| Process noise $Q_0$ (*deg*) | 0.001 | 0.001 | 0.001 |
| Measurement noise $R_0$ (*deg*) | 0.001 | 0.001 | 0.001 |
| Adjustable parameter $\delta_j$ | 1 | 1 | 1 |
| Random process noise $\omega$ (*deg*) | 0.05 | 0.05 | 0.01 |
| Adjustable parameter $a_j$ | 0.8 | 0.8 | 0.8 |
| Adjustable parameter $b_j$ | 0.1 | 0.1 | 0.1 |
| Adjustable parameter $c_j$ | 0.1 | 0.1 | 0.1 |

The output of the three-axis encoder is used as the reference of angular attitude through coordinate transformation. The coordinate transformation from body coordinate system to geographical coordinate system can be decomposed into three basic rotations around Z-axis, X-axis, and Y-axis, respectively, and each basic rotation corresponds to a transformation matrix. The transformation matrix corresponding to rotation around Z-axis, X-axis and Y-axis are shown in Equation (29), respectively.

$$A_{z,\psi} = \begin{bmatrix} \cos\psi & \sin\psi & 0 \\ -\sin\psi & \cos\psi & 0 \\ 0 & 0 & 1 \end{bmatrix}, A_{x,\gamma} = \begin{bmatrix} 1 & 0 & 0 \\ 0 & \cos\gamma & -\sin\gamma \\ 0 & \sin\gamma & \cos\gamma \end{bmatrix}, A_{y,\theta} = \begin{bmatrix} \cos\theta & 0 & \sin\theta \\ 0 & 1 & 0 \\ -\sin\theta & 0 & \cos\theta \end{bmatrix}. \quad (29)$$

The output data of the motor encoders rotating around the ZXY-axis are defined as $\phi_1, \phi_2, \phi_3$. According to the mechanical structure of the experimental device, define

$$\begin{bmatrix} a_{z1} & a_{z2} & a_{z3} \end{bmatrix}^T = A_{z,\psi} \cdot \begin{bmatrix} 0 & 0 & \phi_1 \end{bmatrix}^T, \quad (30)$$

$$\begin{bmatrix} b_{x1} & b_{x2} & b_{x3} \end{bmatrix}^T = A_{z,\psi} \cdot A_{x,\gamma} \cdot A_{y,60°} \cdot \begin{bmatrix} \phi_2 & 0 & 0 \end{bmatrix}^T, \quad (31)$$

$$\begin{bmatrix} c_{y1} & c_{y2} & c_{y3} \end{bmatrix}^T = A_{z,\psi} \cdot A_{x,\gamma} \cdot A_{y,60°} \cdot A_{y,\theta} \cdot \begin{bmatrix} 0 & \phi_3 & 0 \end{bmatrix}^T. \tag{32}$$

The 3-axis angular attitude in the geographic coordinate system is

$$\begin{bmatrix} \gamma & \theta & \psi \end{bmatrix}^T = \begin{bmatrix} a_1 + b_1 + c_1 & a_2 + b_2 + c_2 & a_3 + b_3 + c_3 \end{bmatrix}^T. \tag{33}$$

In this study, RMSE calculated by Equation (27) is applied to compare the accuracy of signal processing algorithms. Where $y_k$ is reference angular attitude calculated by encoder, $\hat{y}_k$ is angular attitude calculated by corresponding algorithm and $m$ is number of sampling points.

4.2.2. Experiment Results

Experiment results include two parts. The first part is static experiment which mainly shows the influence of gyro drift and maneuvering acceleration interference on attitude estimation. The second part is dynamic experiment which mainly shows the advantages of the proposed method in gyro drift and maneuvering acceleration interference restraint and phase compensation. In the experiment, the motion carrier switched between moving state and stationary state and the 3-axis angular attitude are tested for 5 min. In order to show the results clearly and limited by page length, the data were collected for 10 to 15 s.

In the static experiment, the STD of the corresponding output of accelerometer is shown in Figure 10a and the angular attitude comparison in roll direction is shown in Figure 10c. The STD of the corresponding output of accelerometer is shown in Figure 10b and the angular attitude comparison in pitch direction is shown in Figure 10d.

It can be seen from the partial enlarged detail of Figure 10a,b that when the maneuvering acceleration exists, the STD of the accelerometer output data changes greatly and fast. As can be seen from the partial enlarged detail of Figure 10c,d, a cumulative error exists in the angular calculated by the gyro due to the influence of zero drift. The roll and pitch angular attitude drift error reaches 1.649° and 0.544° in 9 s, respectively. Due to the adverse effect of maneuvering acceleration on the output signal of accelerometer, there is an obvious error in the angular attitude calculated by DFBGA. The maximum error value of roll and pitch angular attitude reaches 54.53° and 19.31°, respectively. The signal with such error will seriously affect the performance of the control system. Consequently, it is necessary to adopt the attitude measurement algorithm to suppress maneuvering acceleration.

In dynamic experiment, the STD of the corresponding output of accelerometer is shown in Figure 11 and the angular attitude comparison in roll direction is shown in Figure 12. The STD of the corresponding output of accelerometer is shown in Figure 13 and the angular attitude comparison in pitch direction is shown in Figure 14. The angular attitude comparison in yaw direction is shown in Figure 15.

The angular attitude calculated by DFBGA and SLAIM has the ability to suppress gyro drift and maneuvering acceleration disturbance and there is no obvious noise. However, phase lag error will be introduced in the attitude estimation process based on Kalman filter. The angular attitude calculated by proposed method can restrain the phase lag and smooth the waveform. The proposed method is robust because it can accurately calculate the angular attitude in the presence of gyro drift noise and maneuvering acceleration disturbance.

Calculate the RMSE between the 3-axis angular attitude calculated by gyro, DFBGA, DFBGA, and SLAIM, the proposed method and the 3-axis angular attitude calculated by encoder according to Equation (28). Compared with the angular attitude calculated by gyro, the RMSE of the angular attitude calculated by DFBGA and SLAIM in roll, pitch, and yaw attitude decreased 26.84%, 33.53%, and 24.23%, respectively. The RMSE of the angular attitude calculated by the proposed method in roll, pitch, and yaw attitude decreased 44.23%, 49.91%, and 46.21%, respectively. Compared with the angular attitude calculated by DFBGA, the RMSE of the angular attitude calculated by DFBGA and SLAIM in roll, pitch attitude decreased 37.69% and 28.53%, respectively. The RMSE of the angular attitude calculated by the proposed method in roll and pitch attitude decreased 52.50% and 86.84%, respectively.

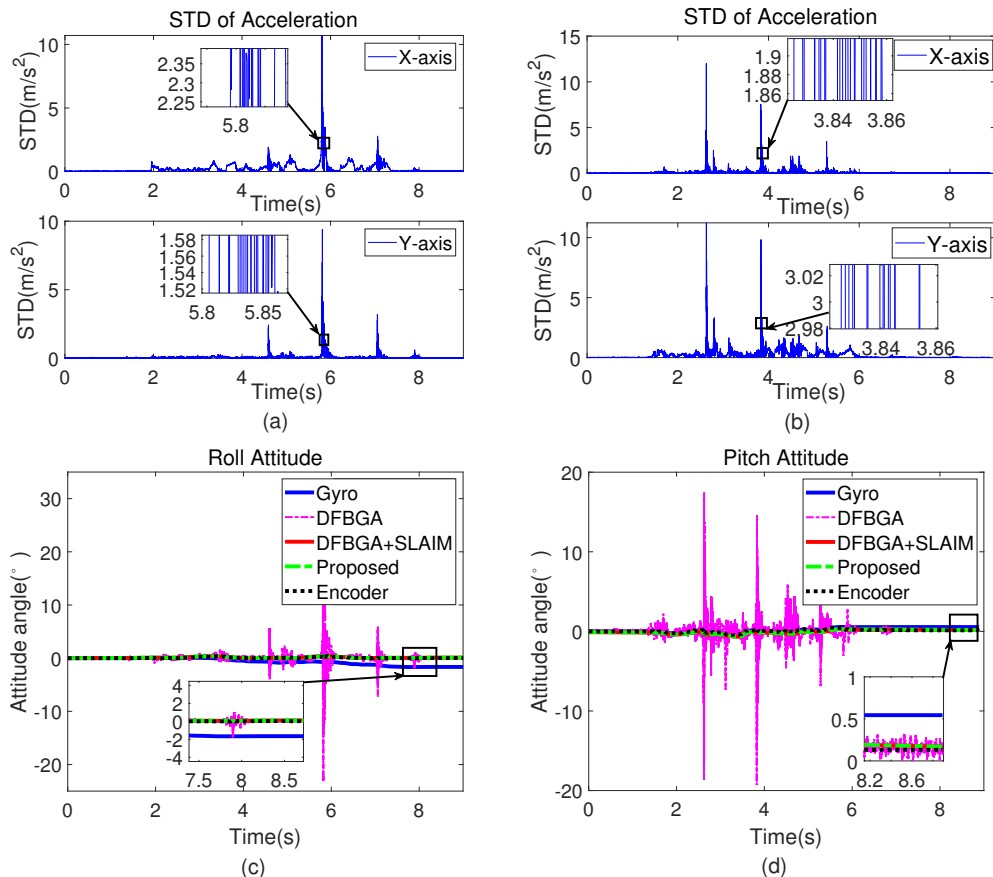

**Figure 10.** Comparison of static experimental results: (**a**) STD of acceleration corresponding to roll attitude. (**b**) STD of acceleration corresponding to pitch attitude. (**c**) Roll angular attitude (calculated by gyro, DFBGA, DFBGA and SLAIM, the proposed method, encoder) comparison. (**d**) Pitch angular attitude (calculated by gyro, DFBGA, DFBGA and SLAIM, the proposed method, encoder) comparison.

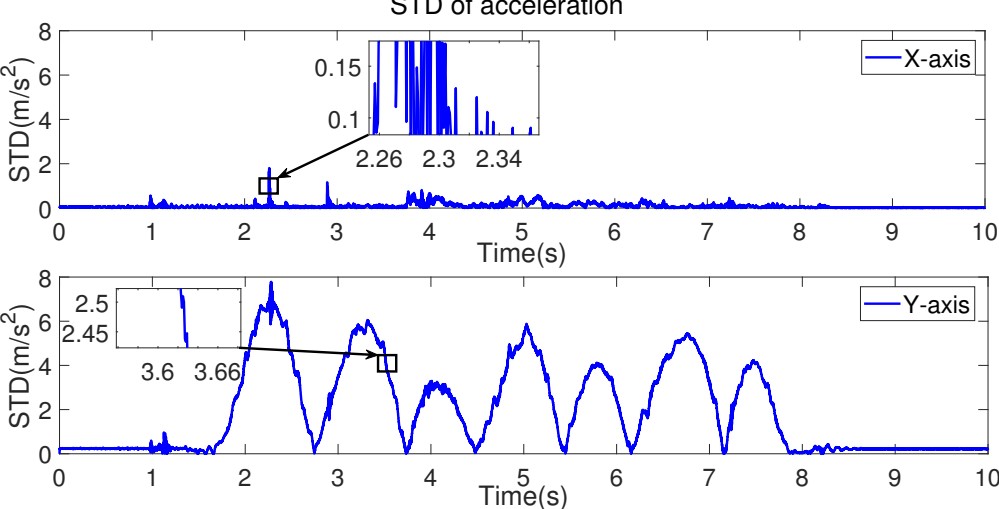

**Figure 11.** STD of X and Y axes acceleration corresponding to roll angle.

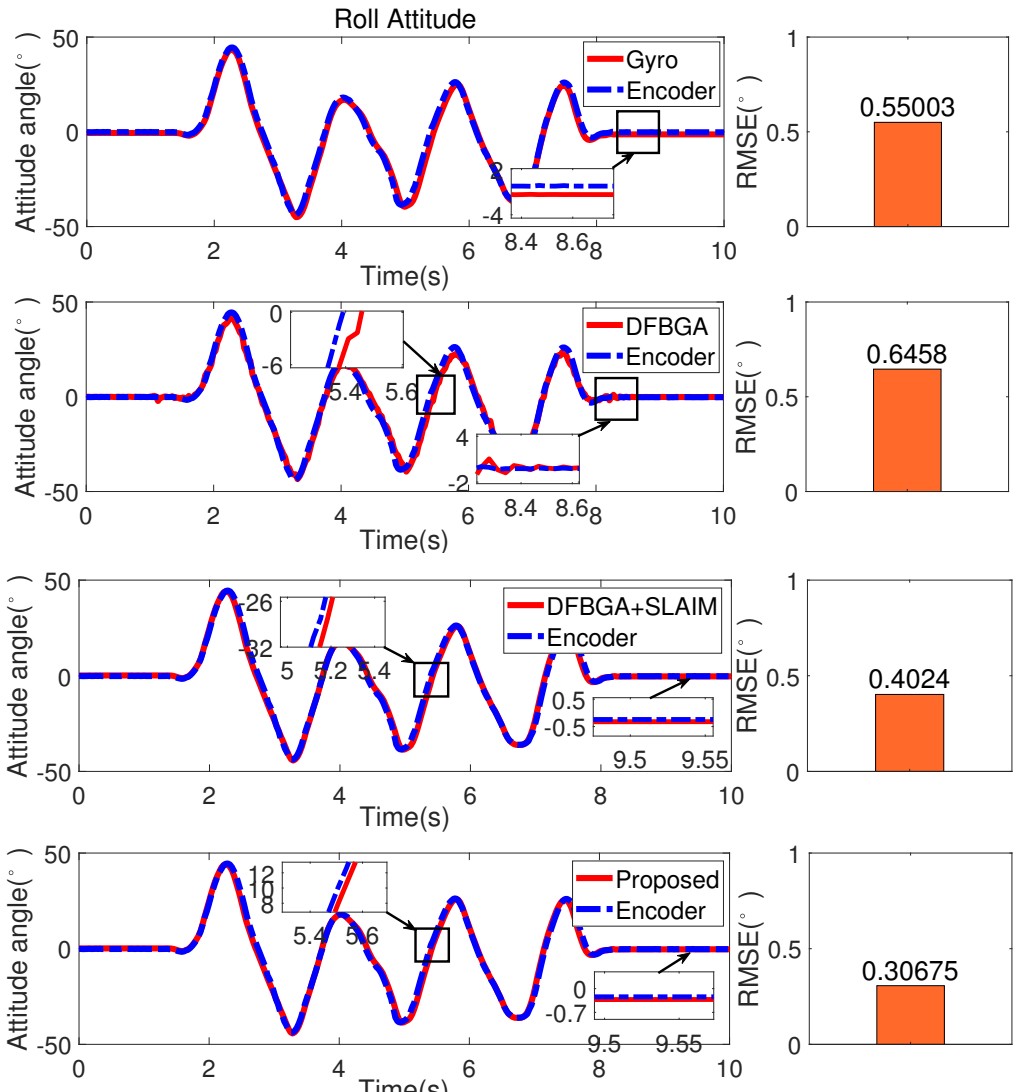

**Figure 12.** Roll angular attitude comparison. The first subfigure is the attitude calculated by gyro, the second subfigure is the attitude processed by DFBGA, the third subfigure is the attitude processed by DFBGA and SLAIM, the fourth subfigure is the attitude processed by proposed method.

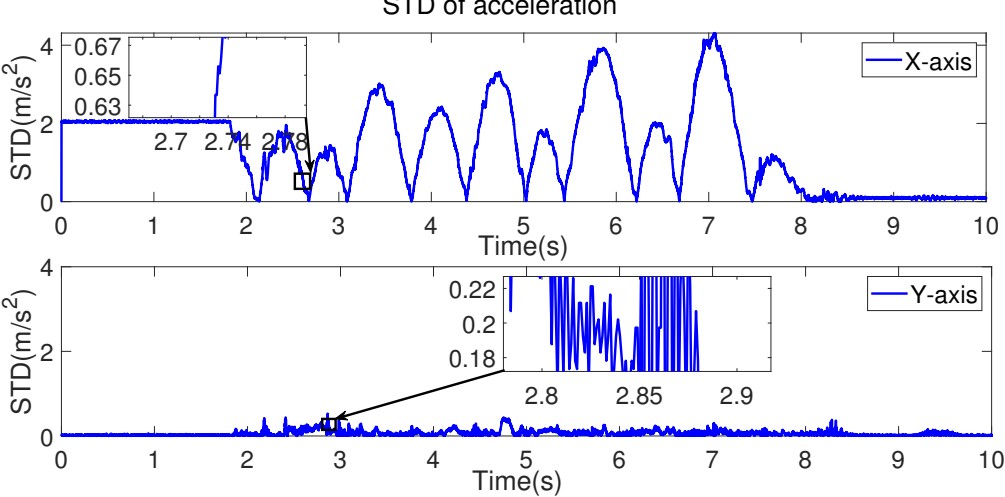

**Figure 13.** STD of X and Y axes acceleration corresponding to pitch angle.

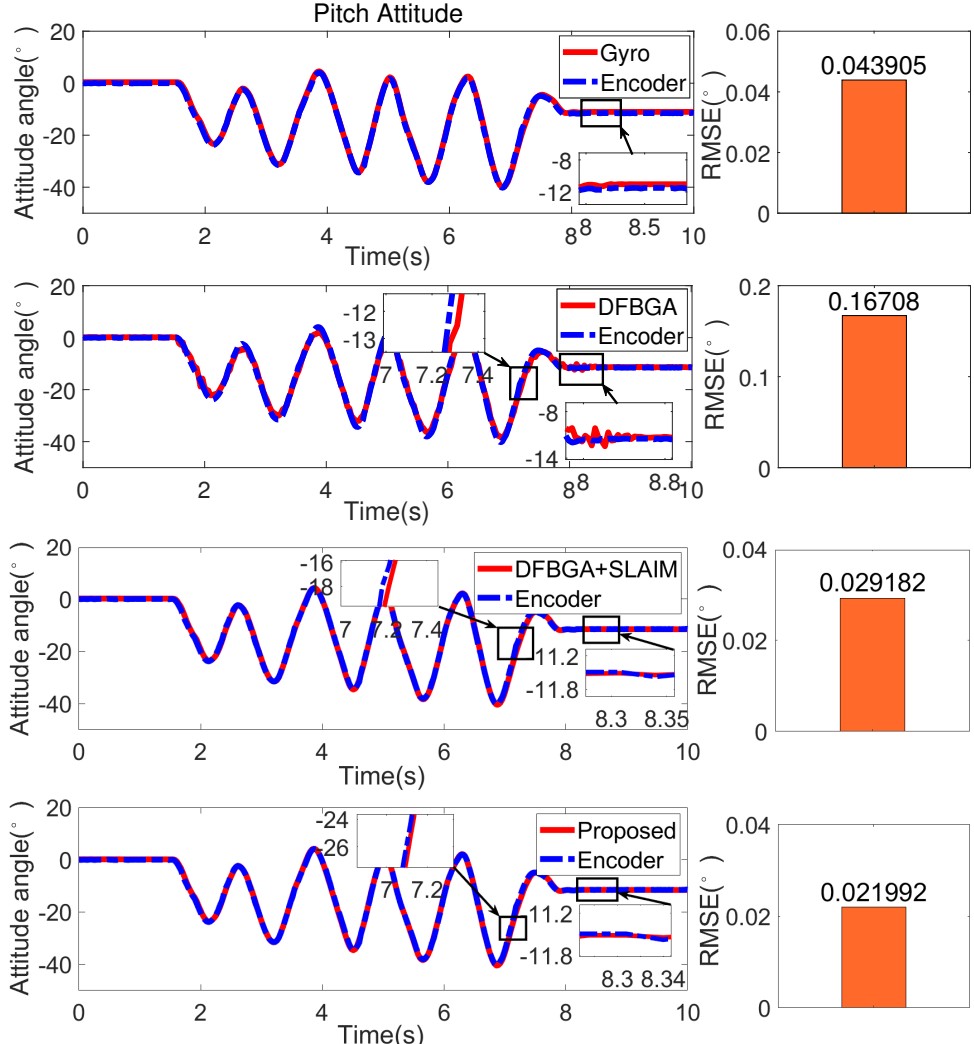

**Figure 14.** Pitch angular attitude comparison. The first subfigure is the attitude calculated by gyro, the second subfigure is the attitude processed by DFBGA, the third subfigure is the attitude processed by DFBGA and SLAIM, the fourth subfigure is the attitude processed by proposed method.

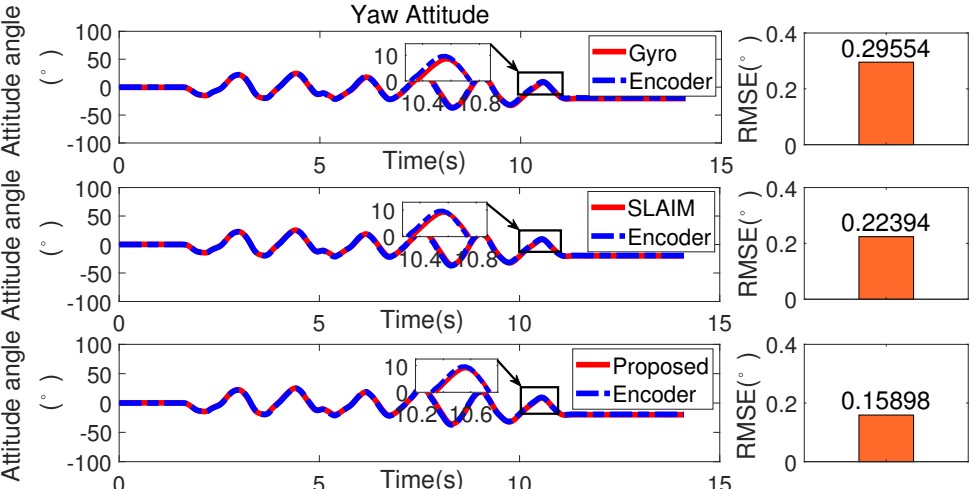

**Figure 15.** Yaw angular attitude comparison. The first subfigure is the attitude calculated by gyro, the second subfigure is the attitude processed by SLAIM, the third subfigure is the attitude processed by proposed method.

Table 6 shows the comparison of time required for data processing once. The proposed method improves the accuracy of sensor data processing at the expense of little processing time. The method proposed in this paper is superior to the traditional processing method in measurement accuracy. Under the gyro drift and maneuver interference, the proposed method can accurately estimate 3-axis angular attitude without phase lag. This high-quality signal can effectively improve the performance of the control system.

**Table 6.** Comparison of time required for data processing once.

|  | Common Fusion Method | Proposed Method |
| --- | --- | --- |
| Roll | 0.4487 ms | 0.5189 ms |
| Pitch | 0.4346 ms | 0.5013 ms |
| Yaw | Unavailable | 0.223 ms |

## 5. Conclusions

In order to improve the attitude measurement accuracy of aerospace optoelectronic platform, an attitude estimation algorithm is introduced in this paper. An adaptive threshold criterion is designed to judge whether the influence of maneuvering acceleration can be ignored. It has good flexibility and versatility because the threshold changes with the sliding of the window. DFBGA and SLAIM are applied in order to compensate for gyro drift error and maneuvering acceleration interference. A phase-lag-free LPF makes use of the phase-lead information in the higher order signal measured by gyro to revise the phase lag introduced by the attitude estimation process based on Kalman filter. The proposed method can accurately calculate the angular attitude in the presence of gyro drift and maneuvering acceleration disturbance. In addition, the phase lag can be compensated and is suitable for attitude estimation of feedback loop in high precision control system. In the simulation and experiment of signal processing, the accuracy and real-time performance of the proposed method are verified by comparing with the reference signal and the attitude measured by encoders. In the simulation of motion control, the position response tracks the position command well. The filters in proposed method does not cause the instability of the control system.

This study improves the performance of LOS motion control system of optoelectronic platform from the perspective of sensor signal processing. The algorithm proposed in this paper does not need to change the mechanical structure design and hardware circuit design of the optoelectronic platform in practical application. This method can be applied to various optoelectronic platforms without increasing the migration cost and has strong practicability. It has the advantages of low cost and strong flexibility. In the future work, the attitude angle calculated in this study will be used as the feedback signal to complete the closed-loop of the control system, which can verify the effectiveness of the proposed method intuitively.

**Author Contributions:** Conceptualization, D.T. and Y.K.; validation, Y.K.; resources, D.T.; data curation, Y.K.; writing—original draft preparation, Y.K.; writing—review and editing, Y.W.; supervision, D.T.; funding acquisition, D.T. All authors have read and agreed to the published version of the manuscript.

**Funding:** This research was funded by the Key Research Program of Frontier Sciences, CAS, grant number ZDBS-LY-JSC044, in part by the National Science Foundation of China, grant number T2122001, in part by the Changchun Science and Technology Development Plan Project, grant number 21SH03, in part by the Innovation Team and Talents Cultivation Program of National Administration of Traditional Chinese Medicine, grant number ZYYCXTD-D-202001.

**Institutional Review Board Statement:** Not applicable.

**Informed Consent Statement:** Not applicable.

**Data Availability Statement:** The data presented in this study are available on request from the corresponding authors.

**Conflicts of Interest:** The authors declare no conflict of interest.

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
