# Peer review of "An Adaptive Three-Axis Attitude Estimation Method Based on Multi-Sensor Fusion for Optoelectronic Platform"

_applsci, doi:10.3390/app12073693_

Round 1
Reviewer 1 Report
See attached file, and also please:
- improve references list
- conclusions should emphasize more of the future developments envisaged by the authors
- correct some capital letters in middle of word

Author Response
Thank you for your valuable comments on our manuscript sincerely. The point-by-point responses to your comments have submitted in the attachment.

Reviewer 2 Report
The paper presents an attitude estimation method for optoelectronic platforms, by fusion between gyrometers and accelerometers with filtering of maneuvering events.
The paper must be corrected by a native English speaker.
The major problems I see for this paper are:
- the lack of novelty : the principle of trying to align the gyrometers to the accelerometers by detecting the calm periods has nothing new and doesn't work very well either. The process is always impaired by undetected slow maneuvers. The only solution is the use of GPS speed measurement which is generally quite good, and fuse all data together in a KF.
- The LOS control has generally two aspects: the stabilization of the sensors to avoid motion blur, and precise pointing in unsupervised surveys. The first is easily achievable with gyros only, and the second is only possible with an active GPS positioning. So I don't really see the point of this work.
- The experimental part of the work is seemingly made in laboratory, with the "maneuvers" probably done by hand, and so not reflecting real situations for an UAV for example.
- The experimental part bring results only on the attitude estimation but the ultimate goal of the work is to use this estimation in a control loop; but the behaviour of this setup is not tested in such a situation, and might reveal itself unstable.
Some more specific remarks :
The line numbering is not complete, especially in part 3.
There are many problems with capitalizations after semi colon.
Line 9: further -> furthermore
Line 10 : campensate -> compensate
Line 23: response is noun, not a verb
Line 37: "the magnetometer exists random error" ???
There are multiple occurrences of the expression "vertical to gravity" : must be changed to "orthogonal to gravity".
There are multiple occurrences of the word "sEquence".
Author Response

(The authors gave the same response as above.)

Reviewer 3 Report
General comments
Congratulation from your work. I think that is very interesting improves the time and the accuracy with your proposed. But I think that the paper must be improved because I see many mistakes and the all caption have insufficient information. And the most important, you must analyse what implies this improvement in the altitude values.
Points to improve in text
Line 21, what is it LOS?
Line 37-39. Are you knowing any paper that support this sentence?
Line 54, could you detail LPF as Low Pass Filter? In line 57 you have done to HPF…
Line 204. What is the accuracy from encoders? That is very important because then you compare the values with this reference eq. 33
Line 205. Are Hz, aren’t?
Please insert the units in Table 1 and 2.
The text from 210 to 213 is the same that from 219 to 223
The figure 8 is introduced in line 241 and figure 7 in line 242. Also, the figure 10 is introduced before the figure 9…. Please correct it! The figures will be introduced in order.
The improvements detailed from 253 to 262 are from your experiment? Because these values don’t appear in the figures 8, 10 and 11. How long are your experiments? In the figures only are shown 10 seconds.
It’s necessary include an example, from real or simulated data, to see the real improves in the altitude values.
The conclusion is shorter, please indicate the general results, the applications, the costs of this improvements in real systems, future steps ……
Mistakes in lines
There is a mistake in many lines. The mistake is (sEquence) that appear in Line 141,159 among others
Line 205 frEquency…..
Line 233 anglular….
Line 238 ConsEquently…
Figures
The figure 3, These values are correct? In maneuvering acceleration of plane the standard deviation changes until 150 m/s2??
In Figure 6, the caption will be more accurate, please insert the information necessary to understand the figure, only reading the caption
In figure 7. In y axes the variation are greater than x axes, but it is difficult to see because the limits from y axes are different. Here you can join the panels in one or use the same limit y axes values. Please increase the description in caption.
In figure 8, 10 and 11 is necessary increase the caption to detail all plots. Also the histograms shown are confused, I think to see the histogram with a width fixed and all the differences found, but only appear in axis x the value 1. I’ll want to see a histogram where the x axis appear the RSME and the y axes the occurrence frequency…..
Author Response

(The authors gave the same response as above.)

Round 2
Reviewer 2 Report
Thank you for your very thorough letter to my comments.
Comments on the response letter:
On the point 1 : There will be always smooth maneuvers that won't be detected by any method not based on external data. For example, your method is not usable on fixed wing UAVs, where acceleration are always perpendicular to the plane of the wings and so considered as vertical, so STDx and STDy will always be very small. That might be the case also for balloons.
It is true that the use of GPS may imply extra maneuvers to fully determine the yaw parameter, but this determination is not necessary to detect maneuvers, and thus it can be used in place of your method with more reliability.
Point 2: You should add the content of your answer in the paper.
Point 3: In absence of real experiments, the use of simulation data realistic of different aerial platforms is necessary to judge the effectiveness of the proposed solution. The best would be of course real experiments on actual UAVs.
Point 4: the use of complex filtering in the feedback of a control loop is always tricky; by reducing the phase lag at low frequency you could induce poles in higher frequencies. These poles would only manifest by instabilities in the control loop, and cannot be seen when the filter is used outside the loop. That why I say that the paper don't bring the proof that this method is usable in the real world.
To summarize, I don't think you have completely satisfactorily answered my concerns for the paper, and I would advise that you delay its publication until you have real experimental data that prove the efficiency of your method, and determine more precisely for which application and type of UAV it is suitable.
Author Response

(The authors gave the same response as above.)

Round 3
Reviewer 2 Report
Thank you for your thorough response.
I understand now that you want to design a opto-electronic platform independent of the UAV and of the type of UAV on which it is installed. This is in my opinion ambitious and difficult, and only real tests on various UAVs and flight conditions and types of survey will prove you successful.
But I understand also that unfortunately sanitary conditions render any real tests impossible, so I think we will have to limit ourselves for now to the simulations you proposed, that seem convincing enough.